# Adopting Safe-by-Design in Science and Engineering Academia: The Soil May Need Tilling

**DOI:** 10.3390/ijerph19042075

**Published:** 2022-02-12

**Authors:** Sam Jan Cees Krouwel, Emma Rianne Dierickx, Sara Heesterbeek, Pim Klaassen

**Affiliations:** 1National Institute for Public Health and the Environment, RIVM, 3720 BA Bilthoven, The Netherlands; emmadierickx@hotmail.com (E.R.D.); sara_heesterbeek@hotmail.com (S.H.); 2Athena Institute, Faculty of Science, Vrije Universiteit, 1081 HV Amsterdam, The Netherlands; p.klaassen@vu.nl

**Keywords:** Safe-by-Design, responsible research and innovation, safe innovation, responsibility, academia, teaching and education

## Abstract

In recent years, Safe-by-Design (SbD) has been launched as a concept that supports science and engineering such that a broad conception of safety is embraced and structurally embedded. The present study explores the extent to which academics in a distinctively relevant subset of science and engineering disciplines are receptive towards the work and teaching practices SbD would arguably imply. Through 29 interviews with researchers in nanotechnology, biotechnology and chemical engineering differences in perceptions of safety, life-cycle thinking and responsibility for safety were explored. Results indicate that although safety is perceived as a paramount topic in scientific practice, its meaning is rigorously demarcated, marking out safety within the work environment. In effect, this creates a limited perceived role responsibility vis-à-vis safety in the production of knowledge and in teaching, with negligible critical consideration of research’s downstream impacts. This is at odds with the adoption of a broader conception of, and responsibility for, safety. The considerations supporting the perceived boundaries demarcating scientific practice are scrutinized. This study suggests that implementing SbD in academia requires systemic changes, the development of new methods, and attention for researchers’ and innovators’ elementary views on the meaning of and responsibility for safety throughout the innovation chain.

## 1. Introduction

The main instrument institutions responsible for the execution or governance of research and innovation (R&I) currently have to warrant the safety of innovations is to demand compliance with established safety rules and regulations. However, R&I always outpace the ability of policymakers to draft novel legislation. This results in a gap between the cutting edge of science and innovation on the one hand and the scope of rules and regulations on the other—a phenomenon referred to as the pacing problem [1]. Combined with the promising potential but inherent uncertainty about impacts of emerging technologies [2,3], policymakers see themselves confronted with a big challenge in the development, maintenance and implementation of regulatory frameworks that are meant to warrant safety of research and innovation. Essentially, it presupposes more a priori knowledge and certainty than is realistic. This concerns both what risks are associated with novel scientific insights and emerging technologies, as well as what can be done to minimize those risks or to minimize the harm associated with them. 

Simultaneously, we find that to address grand societal challenges presented by, for example, the pollution, climate change or healthcare needs of growing and aging populations, research and technological innovation in emerging technology fields—such as synthetic biology and nanotechnology—is invaluable [4,5]. Thus, the question arises, how do we responsibly govern intrinsically uncertain research and innovation? This presents us with a classic Collingridge Dilemma: the quality of the knowledge about future impacts of technological innovations is inversely related with the flexibility of the developmental trajectory of technological innovation [6]. Either we resolve any problematic impacts once they become apparent at an advanced stage of implementation, which, as historical examples spanning multiple domains of human activity show, may come at the price of severe health and environmental consequences [7]. Conversely, we could try to prepare well for novel and potentially grave safety issues in an early stage without certainty about whether and how such issues will present themselves.

To address this dilemma and the pacing problem adequately, the distinct approach of Safe-by-Design (SbD) to both govern and practice R&I increasingly gains attention [8,9]. For researchers and innovators to practice SbD means that from the very first stages of research onwards they adjust their working routines and integrate safety considerations. At the same time, they embrace a broad view on what possible safety issues might ensue and anticipate future impacts, ideally through inclusive deliberation processes that support the identification of potentially pertinent risks. A foundational assumption underlying the notion of SbD is that many choices affecting the eventual impact of innovations are made during early phases of research and innovation [10], while the actual impacts are best foreseen by those closest to the societal context in which an innovation lands and becomes entrenched. This means that scientists, engineers, and designers immediately responsible for the research and innovation processes are best situated to practically address any potential safety issue [11,12]. However, identifying those issues should include actors spanning all places and phases that matter to innovation trajectories—hence including regulators and other (in)direct stakeholders. 

Although the concept is still under development and open to multiple interpretations [13], SbD appears to hold the potential to provide a common anchor for the deliberation on and promotion of safety in innovation across different technological spheres, populations, and phases of innovation [14]. Although a growing body of research is emerging on the scope and practical implications of SbD in nanotechnology, biotechnology, and engineering [8,9,15], it is still largely unclear as to what extent researchers in these fields are willing and able to adopt the concept. The objective of this study, then, is to gain insight into the degree to which science and engineering in the Netherlands are receptive towards SbD, what meaning academic researchers and educators attribute to safety in their work and how they perceive their responsibility for safety and for teaching about safety.

In the next section we present a framework to describe SbD and its research context. Several sensitizing concepts are used to analyze the data, gathered through in-depth semi-structured interviews with 29 academics from the fields of chemistry, nanotechnology, and biotechnology, working at Dutch universities. In the results section we show that views on safety can be organized along two axes. The first concerns the distance between the respondents and the potentially relevant safety issue, the second the degree of uncertainty characterizing the issue. Perceptions around responsibility for different types of safety issues and respondents’ views on the place of safety in education are subsequently dealt with. In the discussion, we critically scrutinize some of the perceived divisions of responsibility we found, and suggest routes of inquiry to further understand both the researchers’ current views and the changes that, arguably, are required for implementing SbD.

## 2. Setting the Conceptual and Institutional Stage

### 2.1. Safe-by-Design: Technical Knowledge, Inclusive Deliberation, and an Open Mind

SbD is a much-encompassing concept with widely dispersed roots. It builds both on an established tradition of safety science and safety engineering, as well as on ideas around the responsible governance of science and innovation developed under the label of Responsible Research and Innovation (RRI). It has clear linkages to approaches to design and science such as inherently safer design, prevention through design, and green chemistry [16,17,18]. However, while these all provide concrete technical design requirements for realizing safer outcomes, these approaches are not specifically geared towards the context of emerging technologies and the large degrees of uncertainty these are associated with. SbD simultaneously aims to facilitate a critical awareness of and action on potential (downstream) safety issues and persistent uncertainty about broader societal and environmental impacts.

Indeed, SbD represents an effort to make safety an active concern throughout the entire innovation process and life cycle of (emerging) technologies and the materials, processes, and products they give rise to. Perceptions of responsibility for risk and safety, and practices in organizing this responsibility, are central to this effort. Simultaneously, SbD also requires that scientists and innovators master the requisite theoretical and practical insight to adjust their work practice and design strategies for safer innovation. This encompasses attention for (I) (degrees of) uncertainty around safety, (II) safety throughout the innovation chain and product life cycle, and (III) a reflective and proactive attitude towards the safety concerns that emerge from (I) and (II). The first two invite an inter- or transdisciplinary approach to safety issues to ensure that attention is dedicated to what a variety of experts and stakeholders conceive to be potentially pertinent safety issues. 

This arguably requires from researchers and educators that they direct attention to implementing a more societally inclusive research and innovation process and display the appropriate accompanying attitude. Particularly, these aspects of SbD show affinity with RRI [14,19], as a governance framework that emphasizes the importance of anticipation, reflection, inclusiveness, and responsiveness in steering research and innovation towards processes and outcomes that are socially desirable, environmentally sustainable, and morally acceptable [20,21,22]. SbD may therefore be seen as an operationalization of RRI, with a focus on the value of safety. It aims to create a mindset supporting researchers and innovators in bridging the gap between the “hard” technical challenges around risk mitigation and the so-called “softer” challenges around the ambiguity and unpredictability of impacts. The technical challenges generally rely on expert insight, whereas the softer challenges require capacities to identify and address issues that also require involvement of broader ranges of societal actors.

As the concept is still developing, a stringently operationalized framework encompassing all SbD elements is not available. Nevertheless, an exploration of perceptions around SbD requires the explication of some sensitizing concepts. Therefore, a preliminary description of the three general elements is given below to indicate what SbD aspires to.

I.Degrees of Uncertainty Around SafetyThis entails a broad conception of safety including both reactively identifying and addressing hazards to prevent undesirable outcomes (safety-I) and proactively focusing on the capacity to adapt to changing conditions and uncertainties and improve the odds of continued desirable outcomes (safety-II) [23]. In addition, it expands its attention to include not just calculable risks, but also concerns with a degree of uncertainty about either exposure or hazard, unknown unknowns, and concerns that emerge from indeterminacy or normative differences [19].II.Safety Throughout the Innovation Chain and Life CycleThis concerns the relevance given to issues occurring throughout the entire life-cycle of the product which, within SbD, is viewed as part of the potential design space when addressing any type of safety issue [19,24]. This would become apparent when one considers a broad variety of options in advancing the safety of an innovation such as: (re)design of the material, the product, the production process, the work environment, the value chain or the social or natural environment in which it will be used. III.Reflective and Proactive AttitudeA particular mindset in which safety beyond one’s own environment is an active concern regardless of one’s position in the innovation chain, ideally addressed in deliberation and collaboration with a range of pertinent actors. On an individual level this requires actors to assume a certain degree of responsibility, or at the very least an active interest, for the safety impacts of their work beyond one’s direct sphere of influence.

### 2.2. Safe-by-Design in Academia: Research, Innovation and Education

Universities present an important environment to develop the potential for SbD and its associated change in perspective on responsibilities for safety. Traditionally, the university is both an institution at which cutting-edge research takes place, as well as one where future innovators, policymakers, and entrepreneurs are educated. Universities’ role in research means that they constitute an emblematic location for testing and developing ideas and practices around SbD. Moreover, their role in education implies that the knowledge, attitudes, and competencies developed therein will influence both what (technological) innovation will take place outside academia, as well as how innovation is done. Arguably, research and innovation cultures across innovation systems are directly influenced by the education researchers and innovators receive at university. Adoption of SbD at universities would thus have a direct effect on the attention to safety given in research and innovation and carry the potential to substantially affect the future knowledge base and frames of reference of a wide variety of pertinent actors. 

Academics at universities tend to combine research and teaching activities. This means that their views around safety and responsibility for safety are likely indicative of the extent to which SbD may meet fertile grounds both in academic research, as well as within the knowledge, skills, and attitudes transferred to students. The sparse evidence available, for instance from a workshop with PhD students investigating nanomaterials, suggests the attention giving to safety as it is conceived within SbD to be quite limited [25]. Therefore, to know what role SbD could play in warranting the safety of research and innovation, it is of essential importance to further explore how researchers in science and engineering disciplines make sense of safety in their work practice and what role they distinguish for themselves both where it comes to realizing safety and teaching about it.

## 3. Materials and Methods

We conducted semi-structured interviews with academics who work both as researchers and as educators in relevant fields of study, i.e., exemplary fields of science and engineering characterized by a large degree of uncertainty as to safety impacts along the research and innovation chain. On the one hand, fields like Machine Learning and Artificial Intelligence were excluded for the ethereal nature of the types of risk these pose (e.g., to democratic governance or equity; see [26]), while on the other hand fields like aerospace or mechanical engineering were excluded because of the well-established risk governance structures already in place there. Because the work of a third category of potentially relevant experts does not revolve around materially designing novel innovations, scholars in fields such as safety science, Science and Technology Studies (STS), and risk governance were also excluded.

The interviews were structured around the sensitizing concepts described above in outlining the basic elements of SbD. The interviews served to investigate what associations, values and affects participants show regarding these concepts. We deliberately took as the starting point of our investigation only loosely defined concepts revolving around safety, uncertainty, and responsibility as sensitizing concepts that can be employed as “measuring rods”, rather than precisely operationalized concepts [27,28]. This facilitated conversations that would provide insight into the “readiness” of respondents to embrace SbD, without imposing on them a certain meaning or perspective on what embracing SbD or taking responsibility for safety would have to entail.

Participants were invited to partake in an interview regarding views on safety in innovation with an attachment mentioning the concept of SbD in relation to the societal challenge it hopes to address. The interviews were structured to discuss several themes: (1) associations with the meaning of safety; (2) perceptions around the concept of SbD; (3) perceptions on responsibility for safety and (4) current scope of/attention for safety within the educational program and institute. Between freely associating on safety and reflecting on SbD, a short explanation was provided that introduced the following three elements: (2a) a broad conception of safety; (2b) attention for safety throughout the entire innovation chain; and (2c) a change in mindset that would encourage innovators to assume responsibility for safety. 

The interviews took place between March and June 2020. They were conducted either in Dutch or English, depending on the researcher and respondent, and lasted between 45 and 90 min. 30 scientists from ten different Dutch universities have been interviewed, of which one was excluded from analysis owing to a scientific background in STS. Due to the coronavirus pandemic and its implications for face-to-face contact, all interviews were conducted online through the participants’ preferred platforms. Participants worked in either biotechnology (N = 8), nanotechnology (N = 8), or the closely related field of (industrial) chemistry (N = 13). Around half of the participants described themselves as operating more on the fundamental (N = 14) side of science, while the other half indicated being more on the applied side (N=15). Their academic positions varied from lecturer (N = 3), associate professor (N = 11), and full professor (N = 15), several of which perform educational leadership roles such as program coordinator or member of an education committee (N = 8). At the start of each interview, verbal consent was requested, after which consent was again asked on-record. 

Interviews were transcribed verbatim and summarized. Member checks were performed by sending summaries to each participant, ensuring that key points were understood correctly [29]. Verbatim transcripts were coded in ATLAS.ti 8 and analyzed through a combination of thematic and open coding. The data was coded along the general themes of perceptions around safety, responsibility for safety and teaching safety indicated by the sensitizing concepts, and a parallel open coding procedure was adopted, grouping and linking codes to identify patterns. Internal validity was increased through researcher triangulation consisting of an initial round of analyses by three researchers who cross-checked their findings. A second round of analysis was done by one of the initial researchers and two additional researchers, who separately and extensively investigated the data with a similar coding strategy to cross-check and deepen the initial analyses. 

## 4. Results

The results will be presented in three sections. First, the explicit and implicit views on safety that emerged from the data are presented, covering both the degrees of uncertainty that are considered, as well as the presence of a life-cycle perspective. Second, perceptions around responsibility for safety within academic practice are described, and finally a short section covers the perceptions around the attention given to safety in current science and engineering curricula.

### 4.1. Making Sense of Safety

Respondents’ views of safety can be analyzed along two interconnected axes. The first has to do with the physical and temporal proximity of the safety issues they encounter in their day-to-day work. The second has to do with the pertinent degrees of uncertainty around the risks at issue.

The respondents’ first associations with safety were commonly described using terms which refer to the researchers’ immediate working environment, whether that is themselves and their colleagues, the lab and its surroundings or general work practice within their field. This suggests that, first and foremost, safety is a matter of concern within one’s daily work environment. Thus, along the axis of considering safety issues in terms of their relative proximity, we found that respondents from all disciplines considered it self-evidently important that safety is seriously taken into consideration. As P15 formulated it: “There’s a lot of potential risks in terms of working at heights, […] fast spaces, […] with high temperature and pressures. So, we are very, let’s say, positively indoctrinated in making safety the first thing for every task here.” Where it comes to such immediate risks or risky situations as described here, all respondents agree that preventing accidents from occurring in the working environment merits substantial attention. This manifests itself in a distinct work ethic, and in conversations supportive of this attitude.

When queried about less immediate and tangible risks than those respondents encounter in their day-to-day research practice, such as those associated with downstream impacts of the researcher’s work, nearly all participants also expressed a broader conception of safety such as P24: “In a way safety and chemistry, that’s a happy marriage, you know. […] safety in terms of how to design products, and how to recycle it. And then how to do that safely.” This broader view prominently features a consideration of the environment. In addition, a variety of people and groups were mentioned, such as potential end-users of some technological innovation, the scientific community, or specific vulnerable groups in society. When referring to people, most respondents emphasized protecting physical health while a few also referred to the impact on people’s mental health or socio-economic well-being. As visualized in Figure 1, although work safety is a primary concern, participants also generally acknowledge the importance of potential downstream impacts: distant safety issues related to situations characterized by larger degrees of uncertainty.

However, while respondents were quick to acknowledge these broader safety concerns and the importance of addressing them, for the majority it did not appear to be a prominent subject of regular or substantial reflection in their daily work or teaching practice. “We make new products; we make new chemicals. Are they sustainable? Are they poisonous? Are they explosive, whatever? That is not directly what we envision or we think of at the moment we do our research. But it is in a broader sense, at a certain moment it becomes important.” (P29). Comments from many respondents suggested they are not used to discussing such a broader context of safety in direct relation to their work, nor to reflect on the potential adverse impacts of their work. This became apparent through explicit comments on the unexpected nature of the interview (“I was confused, because you dropped me a question: safety in innovation. I do not know what that means.” (R8)) or through reactions indicating respondents being surprised or unsure about how to reply to the questions. Often this was accompanied by comments such as P9’s: “The lab perspective on safety is the closest with what you would consider the safety of the manufacturing process, the people who make the material, that they are protected. […] What happens with it when it goes out of the factory, out of the lab? This is something I had to learn in the last few years. That is something that does not come natural to a scientist”. This broader conception requires perceiving safety in a different sense, indicating an explicit distinction, visualized in Figure 2, between safety within the workplace and safety of the work being done for people, the environment, and its future implementation. 

As far as the broader perspective on safety surfaces, many respondents seem to implicitly distinguish between issues with a potential for immediate action on one side and more distanced systemic issues on the other. The level of immediate action becomes apparent through the discussion of impacts on people or environment and the means to address those impacts through some measure of hazard or exposure reduction. At the systemic level views are shared concerning the roles and impacts of science and innovation in society, as are normative statements on the direction that research in general or that individuals working at different places within the research, education, and innovation system should take to facilitate a safer outcome. Follow-up questions on the relevance of these issues to the discussion on safety evoke several participants to state that they do not consider the latter elements to be part of what it means to reflect on or deal with safety in their work. Implicating another boundary as visualized in Figure 3, they associate these issues with adjacent but distinct fields of research, such as ethics or philosophy of science.

These results indicate, first of all, that respondents make an explicit distinction between safety concerns directly relevant to their work environment and those relevant to the downstream impact of their work, suggesting that safety is not generally viewed from the vantage of a product life-cycle. Although many safety measures discussed were indicative of a reactive approach, the data did not clearly show whether respondents also take a proactive view on safety. It is clear, however, that the types of uncertainties considered are primarily those with a calculable or at least identifiable adverse impact, while those with a higher degree of uncertainty borne out of unknowns or unpredictable interactions are not seen as part of a concern for safety. If at all to be controlled: “You cannot regulate that because it’s part of things that are not known yet, so regulations don’t help there.” (R26). Although this already gives some indication of the scope of responsibilities that respondents perceive around safety, a deeper look at responsibility for safety in the following section provides additional insight.

### 4.2. Responsibility for Safety

Once the conversation turns to taking responsibility for safety, participants often delineate their sphere of influence and responsibility: “So for me it’s easier to do my job properly. To do things that I am good at, right? To make sure that my immediate surrounding is safe and that I am not harming anyone. And the rest I cannot easily control.” (R28). The phrasing used on this topic seems to indicate that responsible behaviour with respect to lab safety and data security are key to the scientist’s professionalism. At the same time, it seems to clearly demarcate the responsibility of the individual scientist to only these concerns related to a safe working environment. In addition, a clear set of rules and responsible actors often define this responsibility: “…if you live up to the regulations that are already in place in the lab, so that’s also making it easier and maybe not necessary for a fundamental scientist to think about safety… Because he or she is already doing that by definition.” (R4). Responsibility for a safe working environment is generally formalized in procedures and protocols, visualized in Figure 4, with a hierarchy that designates people who make the rules, those who follow them, and those who see to it that they are upheld. For some, such as R4 in the example, these safety procedures make it possible to completely externalize a concern for safety. Consequently, a critical attitude towards safety and its broader implementation appears to be unnecessary. Regular comments on how safety procedures are considered boring or obstructive, and how scientists need to be reminded of the value of these procedures and the ways these are enforced, strengthen the impression that formalization does not contribute to an inherently critical attitude towards safety.

Towards the scientific community or society as a whole responsibility of individual scientists mostly appears to concern other values than safety: “…we have some responsibility. Responsibility in the sense that the operational principle of our invention should be robust and foolproof. And in that sense: safe.” (R10). Participants refer to a duty to provide robust, replicable, and useable scientific results through ethically acceptable methods and by avoiding “sloppy science”. Additionally, there is recurrent mention of communicating these results, including potential safety concerns, in a transparent and accessible manner. Responsibility towards society is primarily framed in terms of rigour, integrity, and transparency with respect to the scientific process and the allocation of public money. For most scientists it also seems to end there, as exemplified by R29: “Of course [new material] might be a no go for application later on, but that’s not our core business. Our core business is doing the research.” This suggests that scientists deliver knowledge that provides a basis for further development, at which point responsibility for safety is implicitly transferred: “I am an innovator, I am a researcher, of course. But I do not sell products. […] it’s up to the suppliers and seller to make sure that only safe products are being sold.” (R17). As visualized in Figure 5, responsibility is handed over to post-academic research and innovation further down the chain, conducted for instance by industry actors who have potentially different interests and will eventually reap the benefits of any commercial product. 

The phrasing above—“*…the rest I cannot easily control*” (R28)—exemplifies the most common explanation for this demarcation and transfer of responsibility. Respondents often indicate downstream impacts of fundamental and even applied science being out of scientists’ sphere of influence: “I assume that if I develop a very good material for an application, all these safety issues of how safe is this for the final user are actually taken by the company which will develop the final device for the consumer.” (R2). And even if it would be within their sphere of influence, some respondents claim an inability to address such concerns. Mostly by referring to the impossibility to deal with issues that are inherently uncertain and unforeseen: “You cannot really always assess the creativity of others, what they will do with your product.” (R18), but sometimes also with reference to their lack of the required types of expertise. R16 for example described how material scientists may be able to flag potential risks of a substance, but that investigating those risks falls to others who are specialized in specific types of risk assessment. In line with this observation another allocation of expertise may be identified by recalling the earlier expression that uncertain impacts belong to a different domain of science. The absence of any mention of processes or methods to address such uncertain impacts in research and innovation such as reflection and anticipation or inclusive collaboration with other experts or stakeholders, strengthens the impression that addressing the systemic and societal safety issues does not fall within the respondents’ expertise either.

A more fundamental issue was raised with respect to the bounds of scientific freedom: “*I think in terms of science, people find things in the lab, which from a scientific point of view are interesting to publish as knowledge, and the scientist should not be too hindered by that [safety regulation]*.” (R18). This respondent argued that while it is of course important for governments to ensure safety through regulation, and that some scientific pursuits are ethically out of bounds, safety concerns should not constrain research too much. This view was shared by many academics, either because the pursuit of knowledge is seen as an end in itself, regardless of the merit of its applications, or because it might result in discarding something beneficial due to early—‘presumptuous’—concerns: “She should not be hindered in creativity and exploring many different molecules, like in drug screening, because of potential safety issues way down the line. She should be free to explore and propose.” (R23). This suggests that respondents view safety concerns as a barrier to the pursuit of knowledge and not something that is aligned with their primary responsibilities. A more serious perceived threat towards scientific freedom follows from an apparent conflation of the notions of responsibility and liability. Even though it was never suggested that assuming responsibility for downstream impacts also means holding researchers accountable, several respondents mentioned it would be highly counterproductive to hold scientists responsible for the negative outcomes of third parties using their insights to create or do something harmful to society. “Obviously they should share proper knowledge, but they don’t develop products. So, they can’t be responsible. I mean typically a company sells products … They are fully responsible for the impact of those product on society.” (R22). This completes the visualization in Figure 6 of the respondents’ perceptions around types of safety issues, pertinent expertise, and allocation of responsibility, with a role for policy actors and regulators in covering downstream impacts.

One respondent stands out in particular by explicitly objecting to the tendency within academia to view one’s professional role exclusively as one of knowledge production, without serious regards for the consequences of the resulting insights. “You cannot say: ‘Yeah, but I did not say you had to use it. I only researched the science of it.’ And that is the kind of mindset that I would like to combat a bit […] They have received the privilege from society to allow them to study very deeply into their trade. To know everything about it. And that also means they have a responsibility towards society, to warn for the things that they can be aware of.” (R20) According to this respondent especially those scientists with a deep technical understanding of a specific subject matter should be considerate of the downstream impacts of their research, precisely because they have received a mandate and the opportunity of society to develop such expertise. Several additional respondents did indicate a similar inclination either by explicitly supporting a broader responsibility of science to society, or by questioning fundamental research performed without due consideration or justification of the safety concerns that a potential application of that fundamental insight might entail. Interestingly, when discussing what students ought to be taught a sligthly broader segment of respondents appears to make suggestions indicative of a need for a different approach.

### 4.3. Teaching Safe-by-Design

In general, the respondents reacted hesitative once the possibility was discussed of educating students in science and engineering for broader appraisals of risk and safety issues. They often raised practical barriers such as limited space in the already challenging curriculum, a concern about educating generalists at the expense of specialists and a low perceived enthusiasm for such ‘softer’ skills amongst science students. “When they have to think out of the box or deal with the social-economic problems this is not so popular. At some point they don’t put enough effort in it and then they bash and they criticize and whatever.” (R21) Respondents regularly mention that safety education suffers a poor image as being boring and bureaucratic. This makes it challenging to engage students and academics alike. In addition, the respondents consistently emphasize that it is essential that students are not only offered theory, but that they also receive concrete and practical options to make their practice safer. Offering such material without practical connection to their scientific work runs the risk of education being perceived by students as an extra demand that is irrelevant to their actual subject.

In line with this the respondents primarily suggest and emphasize averting concrete risks and hazards. But increasingly, a real-life perspective finds its way into safety education: “Safety by design [sic] is something that you also, how should I say it, learn the hard way. Not just from what happens in your immediate environment, but also from what happens in the rest of the world.” (R14) Explicit mentions of a need for a more adequate and realistic understanding of safety mostly referred to evaluating past mistakes and disasters and teaching students how to avoid similar mistakes by making them and learning from it: “What you learn from the most, tragic as it is, is the mistakes that you make in your life. So, we shouldn’t move towards a situation in which students can never make a small mistake.” (R14) While naturally relevant such a reactive approach is less suitable to addressing the less quantifiable uncertainties that SbD aims to address, such as unknown unknowns, and concerns borne out of the interaction with society. These require more reflexive modes of thinking which, according to these respondents, is rather unpopular with their students.

Nevertheless, while only a few respondents appeared to feel responsibility for broader impacts of safety, slightly more respondents do suggest teaching a broader responsibility for safety to students. “That they realize that their responsibility goes further than just the small assignments they get from their boss and their company, but also that you have a responsibility towards society. And the skills for that, you should be able to do creative safety assessments. So not just following the step-by-step procedure, but also use your own creativity, your own imagination to brainstorm together with colleagues what potentially could go wrong and how you could reduce that.” (R11) Respondents suggest a variety of teaching objectives such as being critical about the scope of (safety) regulation, creatively placing your work in a broader real-world context, being considerate of the entire system in which innovations eventually effectuate, the various actors the innovation may be affected by or have impact on, and being able to work multidisciplinary with actors from different sectors. However, rather than a particular skillset, several of the most supportive respondents referred to this mode of operating as a certain mindset. “What you need is a mindset where you take five steps back [observe it from a distance] to look what you are doing within a much broader context.” (R6) Although these suggestions are more in line with the notions around what is neccesary for SbD, the way they were brought up often conveyed a tentative conviction.

## 5. Conclusions

This study explored the meaning that researchers in science and engineering fields attribute to different elements of SbD in order to assess the extent to which the concept currently is or may in the future be adopted in academic research and teaching. Our main conclusion is that the academics participating in this study, with few exceptions, display a very limited openness to the implications of SbD. Support for this conclusion can be drawn out along four lines. Three immediately pertain to the questions we set out to answer, and one ties these three together. To succinctly enumerate them:With respect to the meaning attributed to safety, there appears to be a clear distinction between safety in the work environment, more distant adverse impacts on health and environment, and safety in situations where impacts display higher levels of uncertainty. None of these appeared to be a topic of substantial critical reflection in academic practice.Concerning perceived responsibility for safety, respondents only see a role for themselves in the context of issues they perceive to be within their immediate sphere of influence, such as ensuring safe working conditions.While presently respondents’ experiences in teaching about safety is largely confined to teaching about formalized procedures pertinent to what happens on the laboratory shop floor, respondents’ perspectives on what future generations of researchers and innovators ought to be taught about safety suggest they do see added value in expanding and articulating safety concepts in ways that go well beyond current practices.

Arguably, what we see here are three instances of boundary work [30]—i.e., three ways in which academics create or reinforce demarcations between what they do and what they do not take to be relevant to them and their own academic practice as researchers and, especially, educators. In the discussion below, we will relate these to the fourth conclusion, which concerns yet another form of boundary work—one that appears to be transversal to all others and helps understand the former:4.Respondents’ cohering perspectives on (i) safety, (ii) responsibility for safety and (iii) teaching about safety fit well with their efforts at demarcating what constitutes good science from what is not.

## 6. Discussion

In this final section, we will briefly reflect on, position and contextualize the conclusions presented in Section 5.

### 6.1. Where Safety Can Be Safely Considered

The first two conclusions are closely connected. Respondents’ views on safety are rather limited. This becomes clear in the predominance of respondents who view safety as being of central importance to their professional work, and simultaneously as something that is extraneous to the actual output their work results in. This is because the topic of safety is rigorously demarcated as being about a safe working environment. Accompanying this fairly narrow conception of safety, most indicate feeling little responsibility for warranting safety outside the scope of their work practice. Schwarz–Plaschg et al. [31] have previously suggested that a limited scope of safety may contribute to a marginalization of broader safety concerns that might arise further down the innovation pipeline. Indeed, the interviews taken together clearly illustrate that these respondents are unfamiliar with relating to safety concerns outside their direct work environment. 

The way respondents tend to describe safety is analogous with findings by Edwards and Jabs [32], which describe how safety culture is generally described at bureaucratic organizations—presented as a shared responsibility but nonetheless highly formalized in rules, regulations, protocols and hierarchies. Although all participants affirm the importance of such procedures, they are also regularly viewed as an administrative burden which reflects findings from explorations on implementing SbD in academic safety regimes [33]. Such attitudes towards safety procedures suggest safety is seen as a preconditional issue in the larger scope of work responsibilities, which may encourage check-list behavior. Although this could potentially support researchers’ and educators’ internalization of standardized safety procedures, Edwards and Jabs [32] showed that it comes with risks of complacency and the diminishment of active, critical reflection. Even if regulation and accompanying procedures demand of researchers that they cover a much broader scope of safety concerns, as for example is often the case with biotechnology regulation, this attitude in effect accommodates researchers having a blind spot for the uncertain impacts, impeding the critical attention for safety as advocated by SbD. 

### 6.2. Teaching and Views on Knowledge

One potential implication this has for teaching practice is suggested by research that demonstrates the role personal epistemologies have on how topics are taught [34,35,36]. Studies around the concept of personal epistemology investigate the extent to which individuals’ beliefs about the nature of knowledge and processes of knowing influence their decision making, actions and, importantly, teaching practices [37]. Empirical evidence indicates that domain specificity can be discerned within personal epistemologies (i.e., depending on the topic one focuses on, individuals can hold different beliefs about the nature of knowing and the value ascribed to knowledge) [38]. This domain specificity becomes apparent here in viewing one’s domain of science as a complex field of study well worth pursuing, while reducing the topic of safety to a series of procedures that need to be followed—rather than seeing it as a topic that merits investigation in its own right. 

The attitude towards safety that we observed in our respondents corresponds to a type of personal epistemology identified by Schraw and Olafson [34] as assuming knowledge to be objective, relatively unchanging and determined by experts. Topics for which such an epistemology is present tend to be taught as relatively static subjects utilizing teacher-centered teaching methods that demand little active reflection by students [34]. Given that there is a high probability that the teacher’s perception of safety is limited to a series of procedures such teaching methods are likely to reproduce uncritical attitudes in students towards safety with the accompanying blind spots for downstream and uncertain impacts. Further research into the intricacies of researchers’ personal epistemologies and the interrelationships hereof with views on safety and (possibly) cohering teaching practices is more than welcome, as in our research we only indirectly got a grasp on respondents’ personal epistemologies. To gain a solid understanding of the aforementioned intricacies would require both a good grip on the question what epistemology or what epistemologies are dominant around safety in different fields, and how each interrelates with safety related teaching practices, for which we assume mixed methods research designs would be most appropriate [39,40].

### 6.3. Tilling the Soil

In order to challenge the type of personal epistemology just discussed, a possibly more fundamental issue presents itself in the a clearly demarcated role responsibility that the majority of respondents subscribe to [41,42]. The responsibilities towards society that are readily assumed by our respondents center around the integrity of the scientific pursuit of knowledge, but this pursuit of knowledge is essentially seen as separate from the impact that it may have on the world. In light of the magnitude of the challenges our society is currently facing and the urgency with which they have to be dealt with [43,44], it is arguably worthwhile to find ways of reshaping this role responsibility. Therefore, it is of value to explore some of the explanations offered for this attitude towards safety. These mostly revolve around the delegation of responsibility for safety issues to other actors in the research & innovation system, either because they have more expertise (other scientific experts), are further down the innovation chain (applied scientists/designers) and/or have more direct (financial) interest or liability (industry). A more fundamental explanation centers around the argument that adopting a concern for safety would actually impede the progress of science, either by curbing the scientist or by curbing curiosity itself. As we will show, none of these arguments can really be sustained.

The argument that the concern for safety belongs to other scientific fields and forms of expertise, as shown in Figure 6, presupposes that readily quantifiable concerns can be addressed by toxicologists and risk assessors, and the more abstract uncertainties by research ethicists or philosophy of science scholars. While toxicology, risk assessment, ethics and science philosophy indeed are each complex scientific endeavors in their own right, we would argue that this rigid distinction between the tasks of each discipline constitutes an unjustified—and unwelcome—appeal to tradition. One that is, as described in the introduction, at the heart of the current state of affairs as an exemplary example of the Collingridge dilemma. The expertise of these scientists is already being called in, but only at a much later stage of innovation when the characteristics and potential applications of new materials, techniques, products, and processes have been largely established and are being subjected to safety evaluations just prior, or more often after their introduction in society. At this point only the most grave and acute safety concerns weigh up against the variety of interests aligned with the progression of innovation, allowing many of the more uncertain, long-term issues to slip through. 

Another explanation for the abdication of responsibility is provided by the argument that downstream actors such as applied scientists and industrial actors have more direct influence and (financial) interest. While this may be the case, it hardly prevents actors from prioritizing safety earlier on in the innovation chain, or from voicing their concerns and seeking the input of relevant experts. Moreover, a study into the implementation of SbD in research identified a value conflict around the notion that industry has both the means and the interests to facilitate safety science, suggesting that while commercial interests may stimulate industry to value safety, it simultaneously disincentivizes the transparent sharing of data about safety [33]. Consistent with this, Bouchaut & Asveld [13] found support throughout the entire value chain for the idea that a bigger responsibility for actors at the first stages of research and development would be apt. Especially technical experts at the cradle of new innovations, such as our respondents, should attempt to anticipate issues early on, so the findings by Bouchaut & Asveld [13] suggest.

Some respondents indicate that too much concern for safety could curb scientific freedom, interpreting this concern as the holding accountable of scientists for impacts they consider to be beyond their reach or anticipatory capabilities. This is a valid concern when responsibility is interpreted to imply accountability or even liability rather than responsiveness [45]. However, responsibility for safety can and should be construed as a proactive ex-ante interest, with accompanying efforts, than as a mechanism of ex-post accountability [46]. Such a proactive attitude does not imply accountability but rather an expansion of the spheres of influence and interest of the scientist. Even if a mechanism could be devised to hold scientists accountable for broader safety issues, studies into the implementation of RRI have shown that legal requirements, much like the formalization of safety protocols, narrow the scope of attention and impede reflection, thinking about long-term consequences, and reaching out to stakeholders [47]. Such a strategy would thus effectively work contrary to the aims of SbD in directing attention to uncertain and unpredictable impacts.

Finally, the idea that attention to safety issues could curb scientific curiosity through the early discarding of promising ideas touches upon the most fundamental issue to be resolved, influencing all the previous perceptions and arguments. Arguably, this idea shows how conceptualizing the significance of safety issues for practicing scientific research and academic education are immediately intertwined with boundary questions about what is and what is not good science. As Gieryn argued: “Scientists have a number of “cultural repertoires” available for constructing ideological self-descriptions, among them Merton’s norms, but also claims to the utility of science for advancing technology, winning wars, or deciding policy in an impartial way” [30] (p. 783). Within the present study, conflicting ideological self-descriptions emerge around whether science is about, for instance, soundness of method, (statistical) reliability of conclusions and fit with existing theories, or whether broader considerations are pertinent for conceptualizing *good science*—for instance including considerations around which safety issues might arise through one’s research and how these might be addressed. In the first case a concern for safety may impede scientific curiosity, in the second it becomes an integral part of a search for knowledge, that ultimately stands to benefit society.

It is understandable that arguments over how research contributes to the safety of future innovations or dedicating time and efforts to safety aspects of issues characterized by high levels of uncertainty are rare among the cultural tropes scientists use in their self-descriptions. For decades scholars have lamented the reward structures in science and innovation for being misaligned with giving importance to such themes [48,49]. Systemic solutions for these mechanisms are sought through a broader and sometimes contentious discussion around how universities adjust to new forms of knowledge production that more directly connects to societal needs [50]. Specifically in support of SbD Bouchaut & Asveld [13] suggested facilitating an open dialogue between academics and actors along the value chain about safety issues and allocation of resources and responsibility, while Ishmaev et al. [33] suggest an inclusive redesign of existing safety regimes. The present study, to the degree its limited focus on academics working in the fields of biotechnology, nanotechnology and chemical engineering allows, would add to this that unhelpful attitudes towards the role of broader concerns like safety within science and engineering practice should be fundamentally challenged. Fortunately, the fact that a minority of respondents question the predominant belief system and explicitly acknowledge the need to reflect on safety in a broader sense as part of scientific practice shows that alternative ideologies can emerge. The way in which these novel ideologies reconstitute scientific pursuits and academic research is a worthwhile topic for further study.

## Figures and Tables

**Figure 1 ijerph-19-02075-f001:**
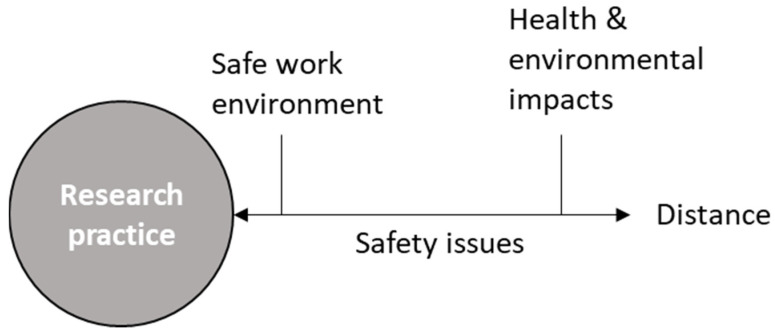
A distinction between types of safety issues close and distant to the respondents’ research practice.

**Figure 2 ijerph-19-02075-f002:**
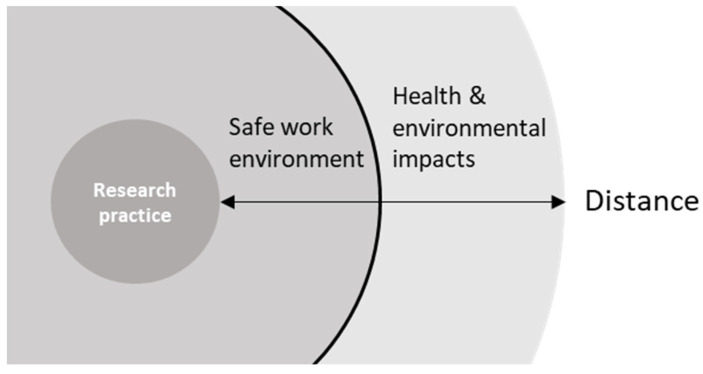
Identification of a boundary between work safety and downstream impacts.

**Figure 3 ijerph-19-02075-f003:**
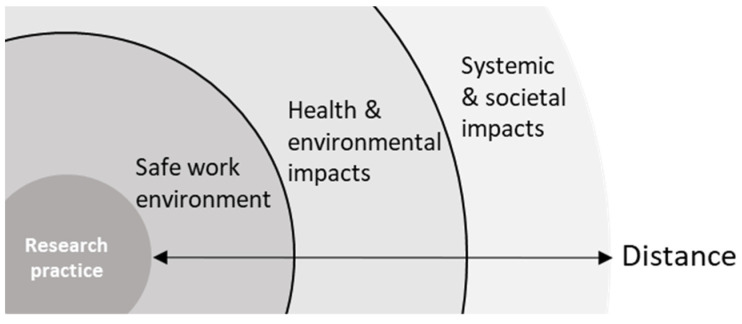
Identification of a second boundary between knowable downstream impacts on health and environment and uncertain systemic and societal downstream impacts.

**Figure 4 ijerph-19-02075-f004:**
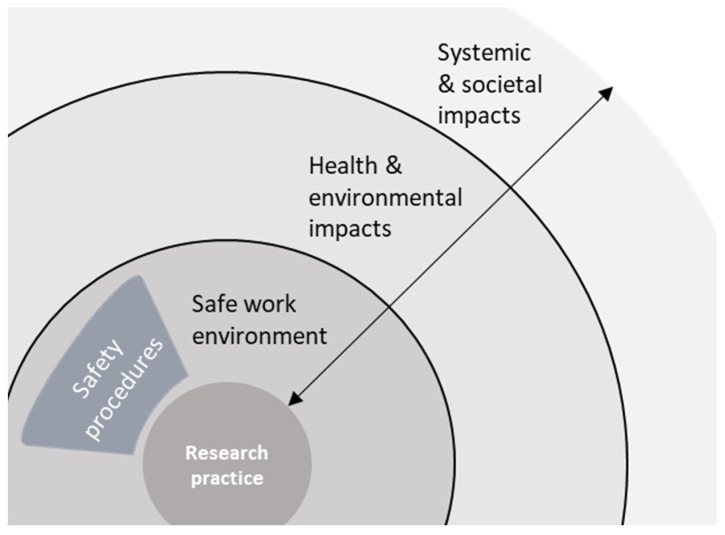
Attention for work safety is covered by adherence to safety procedures (the arrow depicts types of safety issues and distance from research practice).

**Figure 5 ijerph-19-02075-f005:**
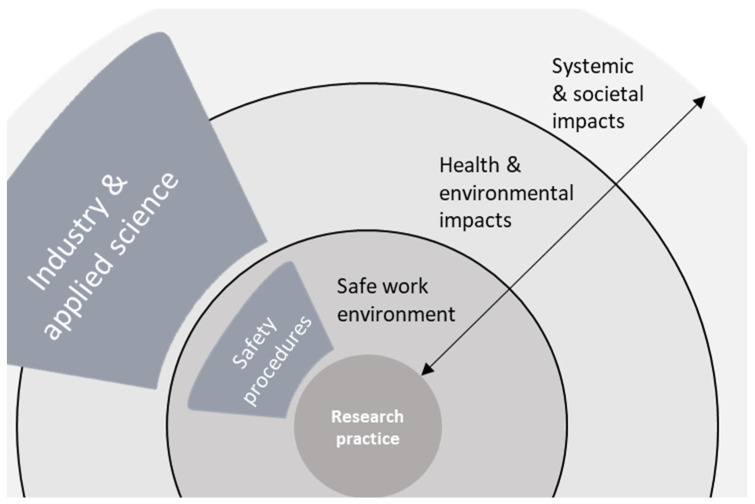
Attention for work safety is covered by adherence to safety procedures. Attention to more distant impacts is or should be covered by industry or applied science (the arrow depicts types of safety issues and distance from research practice).

**Figure 6 ijerph-19-02075-f006:**
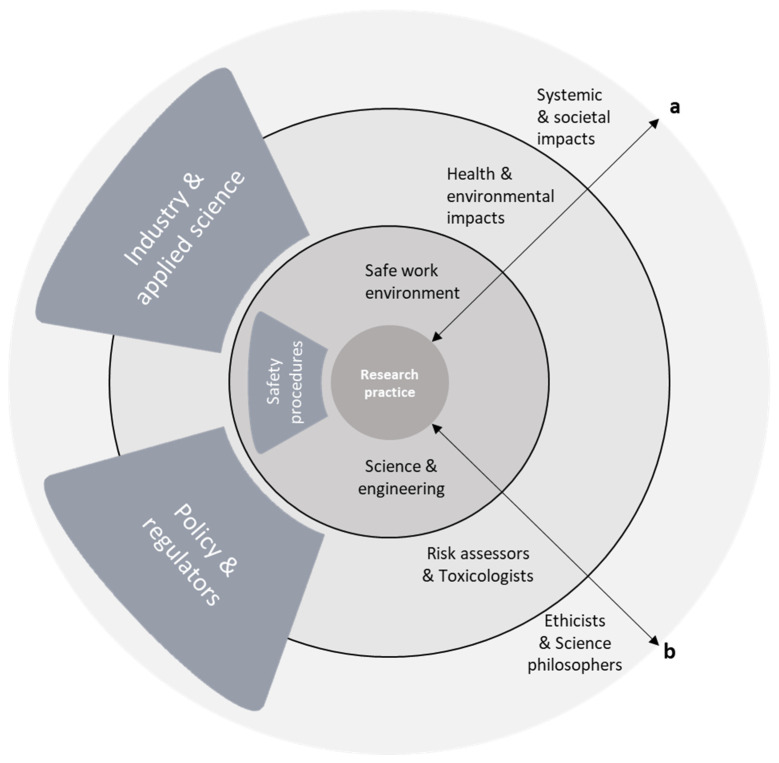
A complete visualization of boundaries, expertise and responsibility allocation around types of safety issues from the perspective of our respondents (arrow a depicts types of safety issues and distance from research practice, arrow b depicts pertinent expertise for increasing degrees of uncertainty).

## Data Availability

The data presented in this study are available on request from the corresponding author. The data are not publicly available due to privacy concerns.

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
