# Peer review of "Adopting Safe-by-Design in Science and Engineering Academia: The Soil May Need Tilling"

_ijerph, 2022, doi:10.3390/ijerph19042075_

Round 1

Reviewer 1 Report

The study is basically well-structured, and I find the figures (workflow) helpful. The topic is clear, however, I miss the scientific soundness. This combined with a quite narrative description of the study I find unlucky for the reader to get quickly the key issues of the paper. I would highlight more the interaction with the interviewed persons. Furthermore, I recommend tightening the whole manuscript. One good example is the last section Conclusions and Discussion: it is an unusual mix which doesn't help too much teh reader. I would separate the Discussion section from the Conclusions one.

Author Response

Response to reviewers of manuscript ijerph-1551404, “Adopting Safe-by-Design in science and engineering academia: the soil may need tilling”

We would like to thank all three reviewers for the time and effort they took to assess the quality of our work and to make suggestions on how we can further improve our article. More specifically, we appreciate the reviewers’ overall positive feedback, and are happy to read that the reviewers consider the paper to be important, relevant, well-structured, understandably written and complemented with helpful figures.

In the table below, we indicate how we have accommodated the distinct criticism provided by reviewer 1.

Reviewer comments

Authors’ response

Reviewer 1

The topic is clear, however, I miss the scientific soundness.

We assume Reviewer 1’s comment on lacking scientific soundness reflects that we come from divergent “epistemic cultures” and/ or disciplinary backgrounds. In response to this comment, then, we wish to point out that ours is an exploratory, qualitative study which first and foremost aspires to reach a comprehension of the perspectives of central actors to the issue at the heart of our research - viz. safety, and responsibility for safety. In the disciplinary context which we draw from, viz. Studies of Science and Technology in Society, a qualitative approach is generally considered to be scientifically sound for dealing with such materials and questions as were ours. We feel we can safely say that the consensus in our field is that the qualitative research approach we took constitutes the ideal means here – i.e., that this is fit for purpose, and hence scientifically sound.

This combined with a quite narrative description of the study I find unlucky for the reader to get quickly the key issues of the paper.

We are sorry to hear that getting to the key issues of the paper was not as easy as it should be. We have made changes to the structure of the introduction that we hope will facilitate readers’ quick understanding of the paper’s key issues. Furthermore, we have slightly revised the abstract to increase clarity and pointedness. Further helpful changes were made in response to the reviewers suggestion to tighten the whole manuscript.

I would highlight more the interaction with the interviewed persons.

Although we appreciate Reviewer 1’s suggestion to highlight the interaction with respondents in more detail, we felt uncertain as to what aspects of those interactions would help in further clarifying our paper’s key issues.

How we have interpreted this comment, then, is as an invitation to provide more detail on why we selected the respondents that we selected. With the more elaborate description of our in- and exclusion criteria of respondents - or rather: of fields from which to recruit respondents - in the methodology section that we have now added, we hope to have accommodated this comment to the satisfaction of Reviewer 1.

I recommend tightening the whole manuscript.

We have tried to tighten the manuscript as much as possible, without compromising completeness and comprehensibility. This has constituted restructuring the introduction, theoretical and contextual background under separate headings. We feel this helps reader arrive at the pertinent questions sooner and has improved overall readability of the paper.

the last section Conclusions and Discussion: it is an unusual mix which doesn't help too much the reader. I would separate the Discussion section from the Conclusions one.

We think this comment is indicative of the great disciplinary variety in literary norms, as in different sub-fields in the social sciences we do think combined Conclusion and discussion sections to be quite common. However, as IJERPH is obviously a highly interdisciplinary journal and comprehensibility for audiences from all pertinent disciplinary backgrounds is certainly vital in our view, we happily accommodated Reviewer 1’s suggestion to separate the Conclusion from the Discussion.

Reviewer 2 Report

General comments

In the manuscript “Adopting Safe-by-Design in science and engineering academia: the soil may need tilling” the authors describe the extent to which academics in science and engineering disciplines are receptive towards the work and teaching practices implied by Safe-by-Design. They also analyse the data, gathered through in semi-structured interviews with 29 academics from the fields of chemistry, nanotechnology and biotechnology.

These topics are of great relevance and this manuscript could provide important reference information that are of considerable interest.

The manuscript setting is well organised and is understandably written; the reader can without great difficulties follow every section The methodological strategy appears correct and clear, fine supported by 50 references.

Note: I have no access to supplementary materials (The following supporting information can be downloaded at: www.mdpi.com/xxx/s1, Figure S1: title; Table S1: title; Video S1: title). Please provide a correct link in the manuscript

Specific comments

I recommend to including a graphical abstract among the figures, both to significantly improving the figures and to facilitating the comprehension of the methodology used.

Author Response

Response to reviewers of manuscript ijerph-1551404, “Adopting Safe-by-Design in science and engineering academia: the soil may need tilling”

We would like to thank all three reviewers for the time and effort they took to assess the quality of our work and to make suggestions on how we can further improve our article. More specifically, we appreciate the reviewers’ overall positive feedback, and are happy to read that the reviewers consider the paper to be important, relevant, well-structured, understandably written and complemented with helpful figures.

In the table below, we indicate how we have accommodated the distinct criticism provided by reviewer 2.

Reviewer comments

Authors’ response

Reviewer 2

Are the methods adequately described?

Can be improved

We have clarified the methods in response to more comprehensive feedback from another reviewer and hope that it is now sufficiently clear.

I recommend to including a graphical abstract among the figures, both to significantly improving the figures and to facilitating the comprehension of the methodology used.

We appreciate Reviewer 2’s recommendation to add a graphical abstract. Unfortunately none of the co-authors have any previous experience with use of this instrument. We have created several versions but our view is that the added value of a graphical abstract is limited for this paper.

This is because while the main figure in the paper does provide a visual chart to illustrate how respondents distinguish various concepts and actors, it does not provide immediate insight in the reasoning behind or concerns about these perceptions. The regular abstract may cover these core findings better than a visual abstract can.

However, since we have little experience with the instrument we have included a version of graphical abstract (see enclosed file). We would like to request input from the reviewer and/or the editor on whether this graphical abstract is of added value and should be included in the paper.

Note: I have no access to supplementary materials (The following supporting information can be downloaded at: www.mdpi.com/xxx/s1, Figure S1: title; Table S1: title; Video S1: title). Please provide a correct link in the manuscript

We assume that this link will be made accessible and corrected in the final publication if and once the paper is accepted and the materials have been published by MPDI.

Reviewer 3 Report

Manuscript refers to the important and current issues of safety, responsibility for safety, as well as safety of future innovation in the context of implementing SbD. It presents the results of on-line interviews conducted among 29 academics from the fields of chemistry, nanotechnology and biotechnology, working at Dutch universities in 2020. The aim of the publication is to determine the extent to which academics are receptive towards the work and teaching practices implied by SbD. The subject of the publication is therefore focused on three groups of issues, which are perceptions around safety, responsibility for safety and teaching safety.

The subject matter was accurately selected, and the described research methodology was clearly presented. The structure of the publication is correct and transparent. The division into chapters is correct.

The publication presents an interesting discussion on the results of the obtained research, questions were posed correctly, and references to the literature were adequately described.

The results of the questionnaire surveys do not raise any doubts as to their correctness, but their content should cause reflection and become a starting point for changes. (e.g. “Indeed, taken together the interviews clearly illustrate that these respondents are unfamiliar with relating to safety concerns outside their direct work environment”).

The obtained responses suggest the conclusion that perhaps also other disciplines scientists, including those directly related to safety, e.g. safety engineering, should have been interviewed. Perhaps a larger group of respondents would have an impact on results too.

So the question arises why was such and not another group of respondents selected? What was the selection criterion and to what extent it influenced the research results.

In view of the above doubts, whether the results can be considered representative, and not related only to representatives of a specific research disciplines?

The added value of the publication could be a more quantitative summary (for example, data presented as a percentage values)

  The publication contained many quotes from the interview. From line 393 there are no italics in the following several statements of the respondents.

Author Response

Response to reviewers of manuscript ijerph-1551404, “Adopting Safe-by-Design in science and engineering academia: the soil may need tilling”

We would like to thank all three reviewers for the time and effort they took to assess the quality of our work and to make suggestions on how we can further improve our article. More specifically, we appreciate the reviewers’ overall positive feedback, and are happy to read that the reviewers consider the paper to be important, relevant, well-structured, understandably written and complemented with helpful figures.

In the table below, we indicate how we have accommodated the distinct criticism provided by reviewer 3.

Reviewer comments

Authors’ response

Reviewer 3

The obtained responses suggest the conclusion that perhaps also other disciplines scientists, including those directly related to safety, e.g. safety engineering, should have been interviewed. Perhaps a larger group of respondents would have an impact on results too.

So the question arises why was such and not another group of respondents selected? What was the selection criterion and to what extent it influenced the research results.

In view of the above doubts, whether the results can be considered representative, and not related only to representatives of a specific research disciplines?

We thank Reviewer 3 for inviting us to explain the in- and exclusion criteria we used for selecting respondents in more detail. In the Methodology section, the fruits of our reply to this comment can be found. We hope our explanation of our reasons for finding precisely thís sample particularly interesting for this first exploratory study is presently sufficiently convincing.

Furthermore, in the Discussion we explain that results may indeed be colored by our sample of respondents.

And finally, please note that the answer to the next comment by Reviewer 3 also speaks to this issue.

The added value of the publication could be a more quantitative summary (for example, data presented as a percentage values).

We kindly disagree with Reviewer 3 about the added value of summarizing the results using quantitative indicators like percentage values, since we are of the opinion that this would suggest a type of precision and a claim to representativeness that we think is neither necessary nor appropriate in the context of the present study. The study is an exploratory one, that for instance looked at what meanings individuals attribute to notions like safety, and how this affects their perceptions of their own responsibility. A carefully delineated yet quite selective sample of respondents was recruited to get to rich descriptions of people’s perspectives, and in the methodology section we do justify the selection of participants and do quantitatively present some relevant characteristics (such as field, and self-identification with fundamental vs. applied science). Where our empirical contribution is concerned, however, the respondents’ point of view is central, and we believe qualitative research constitutes an appropriate approach for getting to this. Although our results could potentially provide an initial empirical and conceptual basis which further research could build on – either quantitatively to test the generalizability of our findings, or qualitatively to reach an even more in-depth understanding of the findings we unearthed with this initial study – this study in and of itself does not aspire to make a generalizable claim per se. Insofar as relative importance or weight of findings is concerned, we trust that the narrative means we employed to “hedge” our claims and indicate their modalities would suffice.

The publication contained many quotes from the interview. From line 393 there are no italics in the following several statements of the respondents.

We have re-formatted so as to make the text consistent throughout in this regard.

Round 2

Reviewer 1 Report

I appreciate the changes done by the authors in response to my queries. I (or better: still) have also the impression that 'we come from divergent “epistemic cultures” and/ or disciplinary backgrounds'. Just as an example: I have never seen a publication with the Conclusions first and finally with Discussions. One have results, one discusses them, and finally one draws conclusions. Sometimes (and I like it) there is an Outlook at the end. In other words, I agree with the revision offered by the authors, I honestly don't fully agree with the general presentation style/language/formulations, but as said it might have to do with the different backgrounds.